# The Structure of the Endophallus Is a New Promising Feature and a Key to Study of Taxonomy of the Subgenus *Metallotimarcha* of the Genus *Timarcha* (Coleoptera, Chrysomelidae) in the Caucasus

**DOI:** 10.3390/insects12100937

**Published:** 2021-10-14

**Authors:** Andrzej Bieńkowski

**Affiliations:** A.N. Severtsov Institute of Ecology and Evolution, Russian Academy of Sciences, 119071 Moscow, Russia; bienkowski@yandex.ru

**Keywords:** taxonomy, morphology, beetles, leaf-beetles, *Timarcha*, *Metallotimarcha*, endophallus, subspecies

## Abstract

**Simple Summary:**

Knowing the correct names of animal species is especially important nowadays; information is posted on the internet, and is available by using keywords. To establish the valid species and subspecies names, it is necessary to research a large amount of material and literature. Of particular importance is the study of type specimens—that is, those specimens for which the names were first established. In the subgenus *Metallotimarcha,* of the leaf-beetle genus *Timarcha*, the external structure of beetles and the shape of the male aedeagus do not allow precise identification of the species. The search for new characters has led to the need to study the internal structure of the male aedeagus. As a result, clear species boundaries of *T. metallica*, *T. corinthia*, and *T. hummelii* have been found. It was established that *T. armeniaca* and *T. hummelii starcki* are the synonyms of *T. hummelii hummelii*.

**Abstract:**

A comparative morphological study of the members of the subgenus *Metallotimarcha* revealed that Caucasian *Timarcha hummelii* is a separate species, which differs from European members of the subgenus (*T. metallica, T. corinthia, T. gibba*) by the internal structure of male aedeagus, namely, manubrium of the endophallus. Morphology of manubrium is described for all species of the subgenus. External characters and the shape of male aedeagus do not permit separating the members of the subgenus distinctly. Examination of both external characters and endophallus structure in the specimens from the Caucasus revealed that *T. armeniaca* and *T. hummelii starcki* are the synonyms of *T. hummelii hummelii*, but not separate species or subspecies. Neotypes of *T. hummelii* and *T. armeniaca* are designated. Syntypes of *T. hummelii starcki* are examined. The key to the species for the subgenus *Metallotimarcha* is compiled.

## 1. Introduction

A new genus-group name *Timarcha* was originally published by Samouelle [1]. The later work by Latreille [2] is sometimes cited as the original description [3,4]. Samouelle [1] proposed the name *Timarcha* without a description, but with the available specific name *Chrysomela tenebricosa* Linnaeus clearly included under it. Thus, the name *Timarcha* Samouelle, 1819 is available [5] (Art. 12.2.5).

The subgenus *Metallotimarcha* Motschulsky, 1860 of the genus *Timarcha* Samouelle 1819 was originally described by Motschulsky [6] as a genus with the type species *Chrysomela metallica* Laicharting, 1781 by the original designation. The type species, *Timarcha metallica*, originally published as *Chrysomela metallica* Laicharting, 1781, is a primary junior homonym of *Chrysomela metallica* DeGeer, 1778 [7,8]. The two species have been included in different genera for a long time. *Chrysomela metallica* Laicharting was combined with the generic name *Timarcha* at least since Herrich-Schaeffer [9], and *Chrysomela metallica* DeGeer was remained in the genus *Chrysomela*, presently *Chrysolina* Motschulsky, 1860. According to [5] (Art. 23.9.5), the case should be referred to the Commission for a ruling under the plenary power; meanwhile, prevailing usage of both names is to be maintained.

To date, the subgenus *Metallotimarcha* includes four valid species, of which two, *T. gibba* (Hagenbach, 1825) and *T. corinthia* Fairmaire and Allard, 1873, inhabit only Western Europe, and one more, *T. metallica*, is widespread in Europe to the Carpathians in the east, and the latter, *T. hummelii* Faldermann, 1837 with the subspecies *starcki* Bechyné, 1953 inhabits the Caucasus and Asia Minor [4]. *Timarcha armeniaca* Faldermann, 1837 is considered a junior synonym for the latter species [4], not being formally synonymized.

The subgenus is characterized by the following features: weak sexual dimorphism: male tarsomeres 1–3 are moderately widened; the marginal border on pronotum is absent; mesosternum is broad and flat; the upper side of the body is metallic; the elytral epipleura are not delimited from above by a border or delimited only at the base; the male aedeagus bears long apical part of tegmen [3,10,11,12,13,14].

All subsequent authors considered *T. hummelii* as a valid species e.g., [3,4,10,13,15,16,17,18,19,20,21,22].

The name “*hummelii*” was used in the original publication by Faldermann [23]. It is correct [5] (Art. 32.2, 32.5). Usage of the name “*hummeli*” [3,10,13,14,16,17,18,19,20,21,22,24,25,26,27] is an incorrect subsequent spelling [5] (Art. 33.4). The correct spelling “*hummelii*” is now predominantly used [28,29,30,31,32,33,34,35]; it was used in a recent well-known catalog of Palaearctic beetles [4] and, therefore, must be preserved [5] (Art. 33.3, 33.3.1).

The name *Timarcha armeniaca* was treated as a species [10,15,17], or as a variation of *T. hummelii* [25], or as an aberration or synonym of *T. hummelii* [4,16,18,21], but was not formally downgraded as a synonym.

Subspecies *T. hummelii starcki* is currently considered a valid one [3,4].

It should be noted that the external morphology in the species of the genus *Timarcha* is variable. Therefore, for a long time, they were looking for new diagnostic characters, in particular, the internal structure of the aedeagus, which was studied in some species [12,36]. Recently, these characters were used to revise the taxonomic position of the certain taxa in the genus *Timarcha* [14,27]. For the species of the subgenus *Metallotimarcha*, the structure of endophallus apodeme in *T. metallica* was studied, and on this basis, differences of the subgenera of the genus *Timarcha* were established [27]. Petitpierre, Anichtchenko [14] examined only the internal sac of endophallus in *T. hummelii*, while endophallus apodeme was not described, and is poorly visible in the published photo. Comparative morphological study of the internal structure of the aedeagus in the species of *Metallotimarcha* has not been carried out until now.

Molecular genetic study, as well as study of the chromosomes of *T. metallica* showed an isolated position of the subgenus *Metallotimarcha* [37,38]. A study of other species of the subgenus has not yet been carried out; it may become the subject of further research.

Larvae of *T. hummelii* and *T. metallica* have been described; they are quite similar to each other and differ well from other *Timarcha* larvae [22,39]

While preparing a key to leaf beetles of the N. Caucasus, the author of the present article paid attention to the difficulty of distinguishing *T. hummelii* inhabiting this region from the European *T. metallica*. To distinguish these species, previous authors [13,17,20] used external features: coloration of the body and legs, body length, the size and shape of the pronotum (position of the greatest width along the length, the shapes of the lateral sides near the base), the presence or absence of the border along the upper edge of the elytral epipleura near the base. However, the author of the present article was unable to find specific differences between specimens from the Caucasus and from Europe by any characters used by the previous authors. A study of the taxonomic position of *T. hummelii* and its intraspecific taxonomy, a comparative study of the morphology of the closely related species *T. metallica*, as well as the internal structure of the aedeagus of other species of the subgenus *Metallotimarcha*, became the subject of the present work.

## 2. Materials and Methods

### 2.1. Material Examined 

Adult beetles were from the following collections: Zoological Institute of Russian Academy of Sciences, St. Petersburg (ZIN), Zoological Museum of Moscow State University (ZMMU), All-Russian Institute of Plant Quarantine, Moscow region (IPQ), Naturhistorisches Museum Wien (NHMW), Zoologische Staatsammlung München (ZSM), Staatliches Museum für Tierkunde Dresden (MTD), All-Russian Institute of Plant Protection (IPP), Museum National d’Histoire Naturelle Paris (MNHN), Hungarian Natural History Museum Budapest (HNHM), Naturhistorische Museum Basel (NMB). In addition, specimens were presented to me by L.N. Medvedev, R.A. Khriapin, E.A. Hatchikov, P.N. Petrov, M.N. Tsurikov, E.V. Iljina, and T.A. Mogilevich. Materials from the author’s collection were also studied. A total of 296 specimens were studied (Table 1).

Regions are grouped according to the territories from which the nominal taxa of the subgenus were described. Detailed information on all examined specimens is included in the Appendix A.

### 2.2. Methods of Examination of Male Endophallus

First, the endophallus structures in all available males was studied. As rightly noted by Daccordi et al. [27], if a preparation of the apodemes is performed, the internal sac of the endophallus is destroyed. Conversely, if the internal sac of the endophallus is everted, the amodemes cannot be fully observed. Attention was placed on the preparation and study of apodemes as more sclerotized and permanent parts that can be prepared more easily to obtain comparable results. In view of the loss of the type specimens of *T. metallica*, the endophallus of the males from the type locality was studied.

Before dissection, the males were found by the dilated tarsomeres 1–3 and the peculiar structure of the last ventrite.

The male specimen was put in water with a drop of detergent for 12–24 h at 20 °C for softing.Elytra were opened, and aedeagus was extracted through abdominal tergites.Aedeagus was soaked in KOH 10% for 12–24 h at 20 °C.Aedeagus was washed in a large amount of water and the apodemes were extracted with a pin hooked at the end through the basal opening of the aedeagus.Apodemes were examined under a stereomicroscope. Then the aedeagus and apodemes were placed in glycerin in a Genitalia Micro Vial GVP 016 microtube pinned under a specimen.

The nomenclature of details of the male genitalia is modified after Petitpierre, Anichtchenko [14], and Daccordi et al. [27] (Figure 1a). The genitalia consists of the median lobe of aedeagus and tegmen, encircling the median lobe. Inside the median lobe, there is an endophallus, consisting of a soft internal sac and sclerotized apodemes. Apodemes consist of flagellum (after Petitpierre, Anichtchenko [14]) (=ductus, after Daccordi et al. [27]) and manubrium (after Petitpierre, Anichtchenko [14]) (=phanera, after Daccordi et al. [27]). Paired wings are attached to the base of the manubrium (these wings are absent in some species).

### 2.3. Methods of Examination of External Morphology and Male Aedeagus

After the examination of endophallus, 12 more morphological characters were studied, including the characters that were used by the authors of the original descriptions of taxa and keys for the species of the subgenus *Metallotimarcha*.

Metric characters (1–4) were studied under a stereomicroscope using a measuring eyepiece. The measurements are shown in Figure 2. Qualitative characters (5–12) were studied under a stereomicroscope by comparison with the “reference” samples, i.e., the specimens with the clearest manifestation of the characters.

### 2.4. Studied Characters

1. Total body length (in lateral view) (Figure 2: 2).

2. Size of the pronotum (a): pronotal length (in dorsal view)/elytral length (in lateral view) (Figure 2: 1, 3).

3. Size of the pronotum (b): pronotal width (in dorsal view)/elytral length (Figure 2: 1, 5).

4. Location of maximal width of the pronotum: distance from anterior margin of the pronotum/total length of the pronotum (both in dorsal view) (Figure 2: 3, 4).

5. Emargination of the pronotal lateral side before the base (1—present, 2—absent).

6. Dorsal color (1—violet, 2—blue, 3—golden-green, 4—golden-coppery, 5—bronze, 6—blackish-bronze).

7. Color of femora (1—rufous, 2—piceous with metallic shine, 3—black with metallic shine).

8. Color of tarsi (1—rufous, 2—piceous with metallic shine, 3—black with metallic shine).

9. Punctures at the elytral disk (five states from fine to large, i.e., approximately 0.007, 0.009, 0.011, 0.013, 0.015 mm wide).

10. Border at the upper margin of the elytral epipleura near the base (1—present, 2—absent).

11. Shine of elytron (1—shining, 2—dull).

12. Shape of aedeagus apex in lateral view (1—recurved dorsally, 2—evenly curved).

Objective methods for recording color morphs, such as colorimetry or spectrometry, may provide better results and may be the subject of future research. However, at present, they are rarely used in taxonomic practice. The simple designation of colors with words prevails in the taxonomic literature and allows comparison of the results of different studies. In the case of the group under consideration, color morph is an additional feature that does not give 100% difference.

### 2.5. Criteria of Subspecies

The author follows the classical rule [40,41], which was recently (successfully) tested by Bieńkowski, Orlova-Bienkowskaja [42]: 97% of specimens in one sample should be separable from 97% of specimens in the other sample, to qualify these samples as representing different subspecies. Amadon [40] showed that this rule is fulfilled for a metric character if the following inequalities are true:|M1 − M2| ≥ 3.24σ1 + 0.68σ2
|M1 − M2| ≥ 3.24σ2 + 0.68σ1

M1 is the mean value of the variable in the first population, M2 is the mean value of the variable in the second population, σ1 is the standard deviation in the first population, and σ2 is the standard deviation in the second population [40]. Subspecies must be defined on diagnosability, not on average differences [43]. We use the classical statistical method for the distinguishing of the subspecies because this is the only method appropriate for distinguishing of the subspecies currently used in zoology, mainly in ornithology and theriology, e.g., [43,44,45,46,47,48]. Unfortunately, the overwhelming majority of insect subspecies are currently described or revised without any statistical treatment, e.g., [14,49,50,51,52].

## 3. Results

### 3.1. Search for the Type Localities and Type Specimens of Timarcha hummelii, T. armeniaca, T. hummelii starcki, and T. metallica

#### 3.1.1. *Timarcha hummelii* and *T. armeniaca*

Faldermann [23] in the second part of “Fauna Entomologica Trans-caucasica” described two species *T. hummelii* and *T. armeniaca*. Type localities were not indicated in the original descriptions. However, in the preface to the first part of “Fauna Entomologica Trans-caucasica”, Faldermann [53] noted that the work was based on the collections of the expeditions by É.P. Ménétries and A.I. Szovitz in the Transcaucasia in 1829−1830.

Faldermann [23] did not indicate the number of specimens studied, but based on their sizes without the limits of variability, it can be assumed that each taxon was described based on the one specimen. The beetles collected during the expeditions by É.P. Ménétries and A.I. Szovitz were deposited mainly in ZIN, some specimens could have been donated to V.I. Motschulsky [54].

After Horn, Kahle [55], types of the taxa described by F. Faldermann, were donated to G.V. Mniszech (that means they can be in MNHN), and deposited in ZIN and ZMMU.

A specimen *T. hummelii* in MNHN is supplied with three historical labels: “Hummelii Fald Fn. TRANSC. II—352. Caucasus”, “Ex Musaeo Mniszech”, and “Museum Paris 1952 Coll. R. Oberthur” (Figure 3c and Figure 4a). Moreover, this specimen bears four recent labels, including two “holotype” labels.

The first historical label refers to the original description, the second part of “Fauna entomologica Trans-caucasica” [23]. The page number is written incorrectly, the description of *T. hummelii* begins with p. 351. The general appearance of the label and the handwriting differ sharply from the original Faldermann labels [55,56]. The G.V. Mniszech collection included the specimens from different collections, donated by different persons, and not only from the Ménétries or Faldermann collections. Due to the absence of original or type labels, labels indicating the type locality, or labels indicating the specimen from Ménétries or the Faldermann collection, it is impossible to definitely attribute this specimen as a type. The specimen in MNHN is a female (according to the structure of the last abdominal sternite and narrow tarsomeres 1–3). Faldermann [23] described clearly a different specimen when he noted for *T. hummelii*: “tarsis <...> valde dilatatis” (=tarsis very wide).

Recent “holotype” labels under the specimen in MNHN are incorrect. The holotype was not designated in the original publication by Faldermann [23]. In this case, all type specimens should be considered as syntypes [5] (Art. 73.1.3).

There were no specimens labelled *T. armeniaca* in MNHN (the curator, A. Mantilleri, personal communication).

In ZIN, the type specimens of *T. hummelii* and *T. armeniaca* were not found. The Motschulsky collection (ZMMU) includes a male labelled “*Timarcha hummeli*” and a female labelled “*Timarcha armeniaca*”. The male *T. hummelii* bears the characteristic Motschulsky collection label with the words “Conf. Persiae” (=at the border of Persia, Iran) (Figure 3a and Figure 4b).

The borders of Russia and Persia, according to the Turkmanchay treaty of 1828, were established in Transcaucasia along the southern borders of the Erivan and Nakhichevan khanates and Azerbaijan [57]. E.P. Ménétries approached this border in Lankaran [54], and A.I. Szovitz crossed the border in Karabakh [58]. Finally, V.I. Motschulsky in 1837 accompanied the Iranian embassy on its way from Russia to the border [59].

Thus, a specimen of *T. hummelii* in the Motschulsky collection could have been collected by Ménétries in Lankaran or by Szovitz in Karabakh and, therefore, could belong to the syntypes, or could have been collected later by Motschulsky near the border of Azerbaijan or Nakhichevan with Iran.

The female *T. armeniaca* bears old label “Achalzik Abas Tuma” (=Akhaltsikhe, Abastuman, villages in S.-W. Georgia) and the characteristic Motschulsky collection label with the locality “Armenia” (Figure 3b and Figure 4c). Ménétries did not visit southwestern Georgia. Szovitz returned to Tbilisi at the end of the expedition and investigated the flora of Mingrelia and Imeretia. Motschulsky visited Akhaltsikhe in the spring of 1835. Thus, this specimen could have been collected by Szovitz (and belong to syntypes), or it could have been collected later by Motschulsky. The indication “Armenia” probably appeared on the collection label as a suggested type locality, based on the specific name “*armeniaca*”.

Both specimens in the Motschulsky collection (ZMMU), *T. hummelii* and *T. armeniaca*, have no type labels or other type markings. They cannot be definitely attributed with the original type specimens. To study the systematic position of these taxa, it is necessary to distinguish neotypes. These specimens are suitable for this purpose. They were definitely collected in the Transcaucasia, the region, by the fauna of which Faldermann [23] worked. These specimens were obviously studied by Motschulsky [6]; they belong to the subgenus *Metallotimarcha* established by him, they correspond to the recent interpretation of *T. hummelii* and the interpretation of *T. armeniaca* as its synonym, thereby ensuring the stability of the nomenclature.

The female from MNHN is not suitable for the designation of neotype of *T. hummelii* for two reasons: (1) for the neotype of the valid taxon, a male is preferable to the female in a group where the main diagnostic features are the structure of the male endophallus. (2) A more detailed type locality is preferably for the taxon, which may contain subspecies, for more accurate understanding of the nominotypical subspecies.

#### 3.1.2. *Timarcha hummelii starcki*

Bechyné [26] described a subspecies *starcki* from the “Western Caucasus” based on an unspecified number of specimens, without a designation of the holotype. Subsequent authors erroneously indicated the Eastern Caucasus as the range of this subspecies [3] or included Armenia in it [4]. In NMB, there are two syntypes of the subspecies *T. hummelii starcki* collected by the famous Russian expert on bark beetles V.N. Starck, with the label “Cauc. Occid. Regio maritima” (=Western Caucasus, seashore region) (Figure 3d and Figure 4d). The Western Caucasus is a part of the Greater Caucasus mountain system, located to the west of the meridional line passing through Mountain Elbrus [60]. Therefore, the type locality is the southern foothills of the Western Caucasus near the Black Sea coast according to [5] (Art. 76A.1.1.). The author of the present study does not designate a lectotype. The designation of a lectotype is justified when there is reason to believe that a type series may include more than one taxon. As a result of the present study of syntypes, it was established that both of them belong to the same taxon. In this case, the designation of the lectotype is optional.

#### 3.1.3. *Timarcha metallica*

This species was described based on an unspecified number of specimens from Tyrol [8]. The J.N. Laicharting collection was deposited in Tiroler Landesmuseen Ferdinandeum, Innsbruck. This collection was lost many years ago [61]. The type of *T. metallica* is missing from the museum (the curator, A. Eckelt, personal communication). There is no need to designate the neotype, since the interpretation of this taxon is clear and unambiguous in recent taxonomical literature [3,13,27,62,63]. If it is needed, for some reason, then a specimen from Tyrol should be designated and it should be placed in the Tiroler Landesmuseen, where the Laicharting collection was originally deposited. The topotypes from Tyrol were examined.

### 3.2. Results of the Comparative Study of the Morphology of Apodemes in Different Species of Metallotimarcha

#### 3.2.1. *Timarcha hummelii*

All studied males from the Caucasus, including *T. hummelii* neotype (Figure 1b,c), *T. hummelii starcki* and *T. armeniaca* topotypes (Figure 1d), have a similar type of apodemes structure. There is insignificant variability in the width of the base of the manubrium, but it has no geographical character, since it is sometimes present in specimens collected in one locality at the same time. 

The manubrium is 1.22–1.42 mm long, well sclerotized, with rudimental wings; the manubrium is narrow, elongate, hardly curved dorso-ventrally, parallel-sided in apical 3/4, broadest in basal ¼, with a semi-transparent medial part at the base, with a narrow furrow consisting of flagellum along with the entire length, ending with two transparent narrow apical lobes. Variability consists of more or less broadened basal ¼ of manubrium. Flagellum is long, narrow, and tube-shaped.

#### 3.2.2. *Timarcha metallica*

Specimens from Western Europe, including *T. metallica* topotypes, and determined by the external features [3,13,16,17,63,64], such as *T. metallica*, have apodemes structures completely different from those in *T. hummelii*.

Manubrium (Figure 1e) is 0.74–0.86 mm long, well sclerotized, without distinct wings, but with paired basal lobes. Manubrium is straight in the lateral view, broadest and quadrangular (in dorsal view) in basal ½, with two triangular basal lobes curved ventrally, narrowest after mid-length, and slightly broadened, bearing irregular denticles along the outer margin at the apex, with narrow furrow consisting of flagellum along the entire length. Flagellum is long, narrow, and tube-shaped.

#### 3.2.3. *Timarcha gibba*

Specimens from Europe, collected near the type locality of *T. gibba* (mountains in the environs of Trieste, partly in Slovenia), and corresponding to the original description [65] and subsequent interpretation of this taxon [3,13,16,17,64], do not differ in the structure of apodemes from *T. metallica*.

Manubrium is 0.85 mm long. Flagellum is long, narrow, and tube-shaped.

#### 3.2.4. *Timarcha corinthia*

*Timarcha corinthia* was originally described from Dalmitia (=Croatia + part of Montenegro). Specimens from Serbia, Bosnia–Herzegovina, Montenegro, corresponded to the original description [10] and subsequent interpretation of *T. corinthia* [3,13,16,17] have apodemes very different from *T. hummelii* and *T. metallica* in the presence of wings at the base of manubrium.

Manubrium (Figure 1f) is 1.71–1.94 mm long, well sclerotized, with long paired basal wings; manubrium is narrow, elongate, hardly curved dorso-ventrally, broadest at the base, and gradually narrowed from the base to apex, with narrow furrow consisting of flagellum along the entire length. Flagellum is long, narrow, and tube-shaped.

### 3.3. Results of the Quantitative Comparison of Samples from Different Regions

Males were identified by the genitalia (manubrium structure). Females, collected at the same time with certain males or in a region from which only one species is known, were identified accordingly. Samples from populations are grouped as follows. Regions 1, 2, and 3 represent the area of *T. hummelii starcki*, 4—the area of *T. armeniaca*, 5—the area of *T. hummelii hummelii*, 6—the total area of *T. armeniaca* + *T. hummelii hummelii*, 7—the area of *T. metallica*, 8—the area of *T. corinthia*.

#### 3.3.1. Total Body Length (in Lateral View)

According to Faldermann [23], *T. hummelii* is 10.16 mm long, and *T. armeniaca* is 13.97 mm long. According to [10], *T. corinthia* is 11 mm long, *T. metallica* is 7–10 mm long, *T. gibba* is 8–9 mm long. According to Marseul [17], *T. corinthia* is “large”, and *T. metallica* and *T. gibba*, both are “small”. According to Weise [16], *T. corinthia* is 10–13, *T. metallica* is 5–10, and *T. hummelii* is 8–13 mm long. According to Bechyné [26], females of *T. hummelii starcki* are 12–13 mm long, longer than those of *T. hummelii hummelii*. According to Medvedev, Shapiro [20], *T. hummelii* is 7.5–13, and *T. metallica* is 5–10 mm long. According to Warchałowski [3], *T. corinthia* is 8.5–13 mm long, but the female is 10.5–15 mm long, *T. metallica* is 6.0–8.5 mm long, but females up to 13 mm long, *T. gibba* is 8–12 mm long, female of *T. hummelii* up to 11 mm long. 

Comparison of *T. corinthia*, *T. metallica*, and *T. hummelii*.

It was found (Table 2 and Table 3) that *T. corinthia* is mostly larger than *T. metallica*, but there is no clear interspecific difference (hiatus) by the length of all specimens, as well as, separately, males and females. *Timarcha hummelii* is mostly larger than *T. metallica*, but there is no species difference too.

Intraspecific variability of *T. hummelii.*

Comparison of regions 4 and 5.

The lengths of all specimens
|M4 − M5| = 0.387916
3.24σ4 + 0.68σ5 = 5.630662
3.24σ5 + 0.68σ4 = 4.578374

Therefore, |M4 − M5| < 3.24σ4 + 0.68σ5 and |M4 − M8| < 3.24σ5 + 0.68σ4.

The difference does not reach the level of subspecies.

The length of females
|M4 − M5| = 0.711477
3.24σ4 + 0.68σ5 = 2.599848
3.24σ5 + 0.68σ4 = 2.270848

Therefore, |M4 − M5| < 3.24σ4 + 0.68σ5 and |M4 − M8| < 3.24σ5 + 0.68σ4.

The difference does not reach the level of subspecies.

The length of males
|M4 − M5| = 0.08601
3.24σ4 + 0.68σ5 = 2.402719
3.24σ5 + 0.68σ4 = 2.098053

Therefore, |M4 − M5| < 3.24σ4 + 0.68σ5 and |M4 − M8| < 3.24σ5 + 0.68σ4.

The difference does not reach the level of subspecies.

Comparison of regions 3 and 6

The length of all specimens.
M6 − M3 = 0.53887
3.24σ6 + 0.68σ3 = 5.212421
3.24σ3 + 0.68σ6 = 4.60373

Therefore, |M6 − M3| < 3.24σ6 + 0.68σ3 and |M9 − M3| < 3.24σ3 + 0.68σ6.

The difference does not reach the level of subspecies.

The length of females
M6 − M3 = 0.929113
3.24σ6 + 0.68σ3 = 2.874378
3.24σ3 + 0.68σ6 = 3.020181

Therefore, |M6 − M3| < 3.24σ6 + 0.68σ3 and |M9 − M3| < 3.24σ3 + 0.68σ6.

The difference does not reach the level of subspecies.

The length of males
M6 − M3 = 0.351852
3.24σ6 + 0.68σ3 = 2.254599
3.24σ3 + 0.68σ6 = 2.120627

Therefore, |M6 − M3| < 3.24σ6 + 0.68σ3 and |M9 − M3| < 3.24σ3 + 0.68σ6.

The difference does not reach the level of subspecies.

Bechyné [26] noted that females of *T. hummelii starcki* are larger than those of *T. hummelii hummelii*. According to the available specimens, the largest female from the Western Caucasus is 12.24 mm long, and the largest female from Transcaucasia not smaller, but even slightly larger, 12.46 mm long. The populations from the WesternCaucasus and Transcaucasia do not correspond to Amadon’s criteria [40] of the subspecies by this character.

#### 3.3.2. Size of the Pronotum (a): Pronotal Length (in Dorsal View)/Elytral Length (in Lateral View) 

According to Weise [16], Medvedev, Shapiro [20], the pronotum is “large” in *T. hummelii*, and “small” in *T. metallica*. However, the authors did not indicate which parameters were taken into account. Two parameters: relative length (present feature) and relative width (next feature) of the pronotum were selected in the present work (Table 4).

Comparison of *T. corinthia*, *T. metallica*, and *T. hummelii*.

*Timarcha corinthia* has, on average, a slightly shorter pronotum, than *T. metallica*, but there is no clear interspecific difference (hiatus) by this character. *Timarcha hummelii* practically does not differ from *T. metallica* in the relative length of the pronotum.

Intraspecific variability of *T. hummelii.*

Comparison of regions 4 and 5
|M4 − M5| = 0.010865
3.24σ4 + 0.68σ5 = 0.169058
3.24σ5 + 0.68σ4 = 0.143778

Therefore, |M4 − M5| < 3.24σ4 + 0.68σ5 and |M4 − M8| < 3.24σ5 + 0.68σ4.

The difference does not reach the level of subspecies.

Comparison of regions 3 and 6
|M6 − M3| = 0.009331
3.24σ6 + 0.68σ3 = 0.160587
3.24σ3 + 0.68σ6 = 0.153314

Therefore, |M6 − M3| < 3.24σ6 + 0.68σ3 and |M9 − M3| < 3.24σ3 + 0.68σ6.

The difference does not reach the level of subspecies.

#### 3.3.3. Size of the Pronotum (b): Pronotal Width (in Dorsal View)/Elytral Length

Size of the pronotum (b): Pronotal Width (in Dorsal View)/Elytral Length (Table 5).

Comparison of *T. corinthia*, *T. metallica*, and *T. hummelii*.

*Timarcha corinthia* has, on average, a narrower pronotum than *T. metallica*, but there is no clear interspecific difference (hiatus). *Timarcha hummelii* has, on average, a slightly narrower pronotum than *T. metallica*. There is no clear interspecific difference (hiatus).

Intraspecific variability of *T. hummelii*

Comparison of regions 4 and 5
|M4 − M5| = 0.00462
3.24σ4 + 0.68σ5 = 0.230477
3.24σ5 + 0.68σ4 = 0.194796

Therefore, |M4 − M5| < 3.24σ4 + 0.68σ5 and |M4 − M8| < 3.24σ5 + 0.68σ4.

The difference does not reach the level of subspecies.

Comparison of regions 3 and 6
|M6 − M3| = 0.016096
3.24σ6 + 0.68σ3 = 0.218365
3.24σ3 + 0.68σ6 = 0.213591

Therefore, |M6 − M3| < 3.24σ6 + 0.68σ3 and |M9 − M3| < 3.24σ3 + 0.68σ6.

The difference does not reach the level of subspecies.

#### 3.3.4. Location of Maximal Width of the Pronotum: Distance from the Level of the Front Corners to the Level of the Greatest Width of the Pronotum/Total Length of the Pronotum (Both in Dorsal View)

According to Faldermann [23], the maximal width of the pronotum is at mid-length in *T. hummelii*, and before the mid-length in *T. armeniaca*. According to [10], the maximal width of the pronotum is at base in *T. gibba*, and before the mid-length in *T. metallica*. According to Weise [16], the maximal width of the pronotum is at base in *T. gibba*, pronotum narrowed anteriorly (slightly more) and posteriorly in *T. corinthia, T. metallica,* and *T. hummelii*. According to Marseul [17], the maximal width of the pronotum is at mid-length in *T. armeniaca*, and before the mid-length in *T. hummelii*. According to Medvedev, Shapiro [20], the maximal width of the pronotum is almost at the anterior margin in *T. hummelii*, and before the mid-length in *T. metallica*. According to Warchałowski [3], the maximal width of the pronotum is at the base in *T. gibba*, at mid-length in *T. corinthia*, and before mid-length in *T. metallica* and *T. hummelii* (Table 6).

Comparison of T. corinthia, T. metallica, and T. hummelii

*Timarcha corinthia* has the greatest width of the pronotum, on average, closer to the apex than *T. metallica*, but there is no clear interspecific difference (hiatus). In *T. hummelii*, the greatest width of the pronotum is on average closer to the apex than in *T. metallica*, but there is no clear interspecific difference (hiatus).

Intraspecific variability of *T. hummelii*

Comparison of regions 4 and 5
|M4 − M5| = 0.02269
3.24σ4 + 0.68σ5 = 0.437507
3.24σ5 + 0.68σ4 =0.379346

Therefore, |M4 − M5| < 3.24σ4 + 0.68σ5 and |M4 − M8| < 3.24σ5 + 0.68σ4.

The difference does not reach the level of subspecies.

Comparison of regions 3 and 6
|M6 − M3| = 0.04741
3.24σ6 + 0.68σ3 = 0.404469
3.24σ3 + 0.68σ6 = 0.341966

Therefore, |M6 − M3| < 3.24σ6 + 0.68σ3 and |M9 − M3| < 3.24σ3 + 0.68σ6.

The difference does not reach the level of the subspecies.

#### 3.3.5. Emargination of Pronotal Lateral Side before Base (Present or Absent)

According to Faldermann [23], the lateral side is slightly emarginate before the base in *T. armeniaca*, and almost without emargination in *T. hummelii*. According to Marseul [17], the lateral side is distinctly emarginate before the base in *T. corinthia* and *T. armeniaca*, slightly emarginate in *T. hummelii*, and without emargination in *T. gibba* and *T. metallica*. According to Medvedev, Shapiro [20], the lateral side is slightly emarginate in *T. hummelii*, and more or less rounded in *T. metallica* (Table 7).

Comparison of *T. corinthia*, *T. metallica*, and *T. hummelii*.

Specimens of *T. corinthia* has emargination more often than *T. metallica* and *T. hummelii*, but there is no clear interspecific difference (hiatus).

Intraspecific variability of *T. hummelii.*

Comparison of regions 4 and 5. Populations from these regions hardly differ in this character.

Comparison of regions 3 and 6. In individuals from the Western Caucasus, the emargination presents more often than in individuals from the Transcaucasia, but the difference does not reach the level of subspecies.

#### 3.3.6. Dorsal Color

Different authors could designate the same colors in different ways. In the present work, a spectrum of colors that reflects variability is adopted. According to Faldermann [23], the dorsal color is purple copper with an elytra dark copper in *T. hummelii*, and copper green with an elytra dark greenish copper in *T. armeniaca*. According to Weise [16], dorsum is brassy in *T. corinthia*, brown with strong brass shine in *T. metallica*, brown, or violet, or greenish, with copper shine in *T. hummelii*, piceous with violet or blue shine in *T. gibba*. According to Marseul [17], the dorsal color is golden bronze in *T. corinthia*, black in *T. gibba*, and bronze brown in *T. metallica*. According to Medvedev, Shapiro [20], the dorsal color is rusty red or purple in *T. hummelii*, and darker, bronze or coppery in *T. metallica*. According to Warchałowski [3], the dorsal color is greenish bronze, copper, blue, or violet in *T. corinthia*, and almost black with blue sheen in *T. metallica* (Table 8).

Comparison of *T. corinthia*, *T. metallica*, and *T. hummelii*.

In *T. corinthia* and *T. metallica*, colors 5 and 6 are sharply predominant, in *T. hummelii,* they are not found, but colors 1 and 4 are predominant. Due to the presence of color types 1–4 in the first two species, there is no clear interspecific difference (hiatus).

Intraspecific variability of *T. hummelii.*

Comparison of regions 4 and 5.

Populations from these regions are very similar in color types. The difference does not reach the level of subspecies.

Comparison of regions 3 and 6.

Populations from the Western Caucasus and from the Transcaucasia are very similar in color types. The difference between them does not reach the level of subspecies.

#### 3.3.7. Color of Femora

Most authors described the coloration of the legs in general. However, since different parts of the leg can be colored differently, the coloring of the femora (present feature) and tarsi (the next feature) were considered separately in the present work.

According to Faldermann [23], legs are brown, almost bronze, with tarsi light in *T. hummelii*, and legs are pitch-brown, slightly copper, with tarsi brown in *T. armeniaca*. According to Weise [16], legs are violet in *T. corinthia*; they are more or less red–brown in *T. hummelii* and *T. metallica*. According to Marseul [17], legs are violet in *T. corinthia*, and blue–black in *T. gibba*. According to Warchałowski [3,13], legs are red–brown or red in *T. metallica* and *T. hummelii*, and black or almost black with metallic shine in *T. corinthia* (Table 9).

Comparison of *T. corinthia*, *T. metallica*, and *T. hummelii*.

*Timarcha corinthia* differs sharply from *T. metallica* and *T. hummelii* in the predominance of color 3, but there is no clear interspecific difference (hiatus). In *T. hummelii* and *T. metallica*, color 2 is predominant; there is no clear interspecific difference (hiatus). 

Intraspecific variability of *T. hummelii.*

Comparison of regions 4 and 5: populations from these regions des not differ by the color.

Comparison of regions 3 and 6.

The proportion of color 2 is slightly higher in region 9 than in region 3, but the difference does not reach the level of subspecies.

#### 3.3.8. Color of Tarsi

Color of Tarsi (Table 10).

Comparison of *T. corinthia*, *T. metallica*, and *T. hummelii*.

*Timarcha corinthia* sharply differs from *T. metallica* and *T. hummelii* in the presence of color 3 in all studied individuals, but due to the presence of this color in a small proportion in *T. hummelii* and *T. metallica*, there is no clear interspecific difference (hiatus).

Intraspecific variability of *T. hummelii.*

Comparison of regions 4 and 5.

Populations from these regions differ slightly. The difference does not reach the level of subspecies.

Comparison of regions 3 and 6.

Populations from these regions differ slightly. The difference does not reach the level of subspecies.

#### 3.3.9. Punctures at the Elytral Disk (5 States from Fine to Large, i.e., Approximately 0.007, 0.009, 0.011. 0.013, 0.015 mm Wide)

According to Faldermann [23], elytral punctures are large in *T. hummelii*, and coarse in *T. armeniaca*. According to Weise [16], elytra are strongly punctate in *T. corinthia*, and, rather densely, more or less strongly punctate in *T. metallica*. According to Marseul [17], elytral punctures are more dense and coarse in *T. gibba*, and more sparse and small in *T. metallica*. According to Bechyné [26], elytral punctures are denser in *T. hummelii starcki* than in *T. hummelii hummelii*. According to Warchałowski [3], elytral punctures are dense in *T. hummelii starcki*, and moderately dense in *T. hummelii hummelii* (Table 11).

During the present study, it was found that the size of the puncture is a more variable parameter than the density.

Comparison of *T. corinthia*, *T. metallica*, and *T. hummelii*.

*Timarcha corinthia* sharply differs from *T. metallica* and *T. hummelii* due to the predominance of puncture state 6; due to the presence of this state in a small proportion in *T. hummelii* and *T. metallica*, there is no clear interspecific difference (hiatus).

Intraspecific variability of *T. hummelii.*

Comparison of regions 4 and 5.

Populations from these regions slightly differ; the difference does not reach the level of the subspecies.

Comparison of regions 3 and 6.

Populations from region 3 are distinguished by the predominance of punctation state 1, which is rare in region 9, but the difference does not reach the level of the subspecies.

#### 3.3.10. Border at Upper Margin of Elytral Epipleura near Base (Present, Absent)

According to Marseul [17], the border is present in *T. corinthia, T. gibba, T. metallica*, and absent in *T. armeniaca* and *T. hummelii* (Table 12).

Comparison of *T. corinthia*, *T. metallica*, and *T. hummelii*.

*Timarcha corinthia* and *T. metallica* differ from *T. hummelii* by the presence of a border in most specimens, but there is no clear interspecific difference (hiatus).

Intraspecific variability of *T. hummelii.*

Comparison of regions 4 and 5.

Populations from these regions does not differ.

Comparison of regions 3 and 6.

About a third of individuals from region 3 have borders, and no individuals from region 6 have borders. Thus, the difference does not reach the level of subspecies.

#### 3.3.11. Shine of Elytron (Shining, Dull)

According to [10], the shine of elytron is less in *T. gibba* than in *T. metallica*. According to Bechyne [26], elytra shines in both sexes in *T. hummelii starcki*, and elytra is dull in females of *T. hummelii hummelii*. According to Warchałowski [3], elytra shines or is dull in *T. metallica*, dull in *T. hummelii hummelii*, and shines in *T. hummelii starcki* (Table 13). 

Comparison of *T. corinthia*, *T. metallica*, and *T. hummelii*.

Shining males predominate in all species, shining females—in *T. corinthia* and *T. metallica*, but there is no clear interspecific difference (hiatus).

Intraspecific variability of *T. hummelii.*

Comparison of regions 4 and 5.

Region 5 has more dull females and males than region 4, but the difference does not reach the level of subspecies.

Comparison of regions 3 and 6.

Region 6 has more dull females and males than region 3, but the difference does not reach the level of subspecies.

#### 3.3.12. Shape of Aedeagus Apex in Lateral View (Recurved Dorsally, Evenly Curved) 

Shape of Aedeagus Apex in Lateral View (Recurved Dorsally, Evenly Curved) (Table 14).

Comparison of *T. corinthia*, *T. metallica*, and *T. hummelii*.

There is no clear interspecific difference (hiatus) by this character. All available males of *T. corinthia* have state 2, as most of *T. hummelii* males, while the majority of *T. metallica* males has state 1.

### 3.4. Descriptions of Neotypes

#### 3.4.1. *Timarcha hummelii* Fadermann, 1837

Male, with label: “Timarcha Hummeli Conf. Persiae” (ZMMU), 9.18 mm long, dorsum golden coppery, femora black with metallic shine, tarsi piceous with metallic shine, tarsomeres 1–3 moderately broadened, entirely pubescent beneath, pronotum without lateral border, elytral epipleura without upper border basally, aedeagus slightly recurved dorsally at apex, apodemes with manubrium long, narrow, elongate, slightly broadened basally, slightly curved dorso-ventrally, wings absent.

#### 3.4.2. *Timarcha armeniaca* Fadermann, 1837

Female, with labels: “Achalzik Abastuman”, “Timarcha armeniaca Fald Armenia” (ZMMU), 10.34 mm long, dorsum golden coppery, femora piceous with metallic shine, tarsi rufous with metallic shine, tarsomeres 1–3 narrow, entirely pubescent beneath, pronotum without lateral border, elytral epipleura without upper border basally.

### 3.5. Key to Species (T. gibba Is Not Included There Because Its Taxonomical Position Is Unclear)

1. Species from Western Europe. Border at upper margin of elytral epipleura near the base mostly present...2.

–Species from the Caucasus and Asia Minor. Border at upper margin of elytral epipleura near the base is usually absent, rarely present. Male genitalia: manubrium 1.22–1.42 mm long, without distinct wings, narrow, elongate. Body 6.86–12.46 mm long. Dorsum usually violet or golden coppery; femora usually piceous, rarely rufous or black, tarsi usually rufous or piceous. The apex of aedeagus is mostly evenly curved...*T. hummelii*.

2. Male genitalia: manubrium 1.71–1.94 mm long, with long paired basal wings, narrow, elongate, broadest at the base, and gradually narrowed from the base to apex. Body 9.08–12.77 mm long. The dorsum is usually bronze, the femora is mostly black, and the tarsi black. The apex of aedeagus is evenly curved...*T. corinthia.*

–Male genitalia: manubrium 0.74–0.86 mm long, without distinct wings, broadest and quadrangular (in dorsal view) in basal ½. Body 6.86–10.50 mm long. The dorsum is usually bronze or blackish bronze, the femora is usually piceous, rarely rufous or black, the tarsi is usually piceous or black. The apex of aedeagus mostly recurved dorsally...*T. metallica.*

## 4. General Discussion and Conclusions

The structure of the male endophallus makes it possible to clearly identify *T. hummelii, T. metallica*, and *T. corinthia.* Most of the external features and the external structures of the aedeagus, including those indicated in the literature, do not allow distinguishing between the species of the subgenus *Metallotimarcha*. Additional features typical for most specimens are given below. *Timarcha corinthia* and *T. hummelii* are mostly larger than *T. metallica.* In *T. corinthia* and *T. metallica*, the body colors are mostly bronze and blackish bronze, while violet and golden coppery colors are predominant in *T. hummelii. Timarcha corinthia* differs sharply from *T. metallica* and *T. hummelii* in the predominance of black in the color of the femora, while the femora are mostly piceous in *T. hummelii* and *T. metallica. Timarcha corinthia* sharply differs from *T. metallica* and *T. hummelii* in the presence of black tarsi in all studied individuals, while this color is present in a small proportion in *T. hummelii* and *T. metallica. Timarcha corinthia* sharply differs from *T. metallica* and *T. hummelii* due to predominance of large elytral punctures, while this state of the character is present in a small proportion in *T. hummelii* and *T. metallica. Timarcha corinthia* and *T. metallica* differ from *T. hummelii* by the presence of a border at the upper margin of elytral epipleura near the base in most specimens. 

We can also take into account the geographical feature: *T. metallica*, *T. corinthia*, and *T. gibba* inhabit Western Europe, and *T. hummelii* occur in the Caucasus and Asia Minor (Figure 5).

*Timarcha gibba* does not differ from *T. metallica* in the structure of endophallus. Taxonomical status of *T. gibba* has been questioned [66]. For a final decision, it is necessary to study additional material.

The manubrium of *T. corinthia* is very different from those in other *Metallotimarcha* members in the presence of wings. This feature is similar to that of *Timarcha* s. str., but the pronotum is not bordered, as typical of the subgenus *Metallotimarcha*. The width and shape of the mesosternum is mentioned as a distinctive feature of the subgenus *Metallotimarcha* [3,10,11,13,14]. According to my results, it does not clearly distinguish between the subgenera.

The endophallus structure (shape of manubrium) in males from the Caucasus has individual variability, but does not give geographic variability. Comparison of individuals from the populations of the West Transcaucasia and East of Transcaucasia (typical areas of *T. armeniaca*, and *T. hummelii*, respectively) and comparisons of individuals from the West Caucasus and Transcaucasia does not give a difference of the sub-specific range. Thus, a synonymy *T. armeniaca = T. hummelii* is confirmed, and a new synonymy: *T. hummelii starcki = T. hummelii hummelii* is established.

The molecular genetic study of the subgenus *Metallotimarcha* may become the subject of further research and will allow testing the conclusions of this article at a new level of knowledge.

## Figures and Tables

**Figure 1 insects-12-00937-f001:**
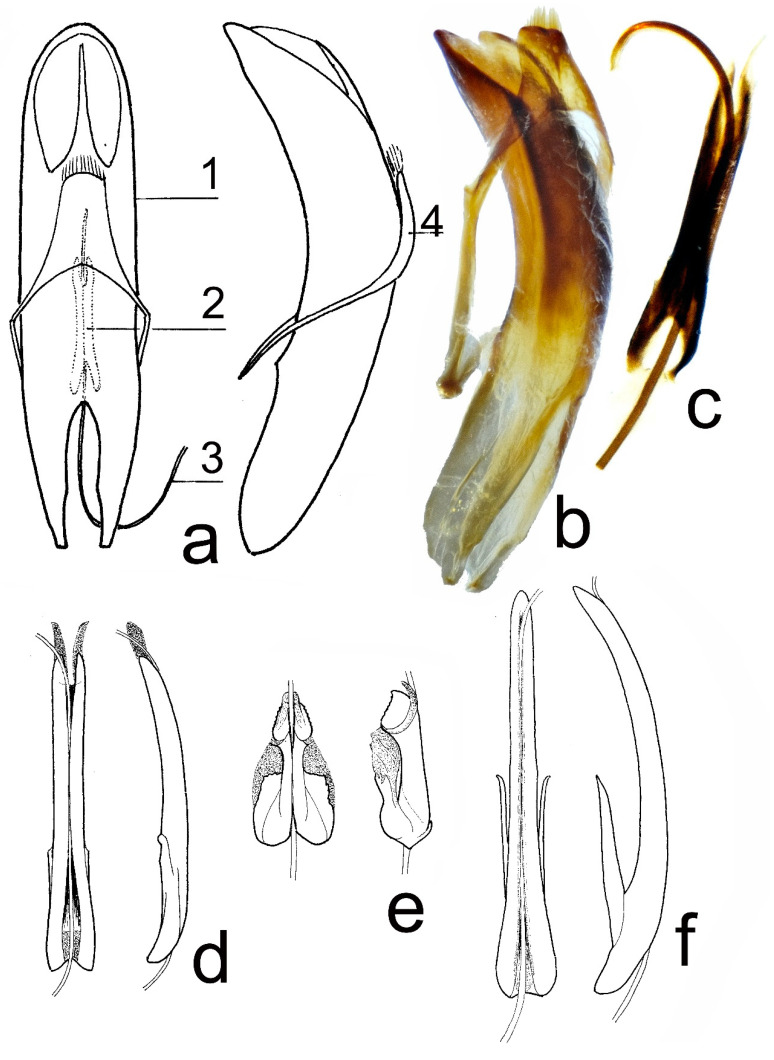
Male genitalia. (**a**) *Timarcha hummelii*, dorsal, and lateral view: 1—median lobe of aedeagus, 2—manubrium within median lobe, 3—flagellum, 4—tegmen; (**b**) *T. hummelii*, neotype, ZMMU, median lobe, lateral view; (**c**) *T. hummelii*, neotype, ZMMU, manubrium, dorsal view; (**d**) *T. hummelii*, W. Caucasus; (**e**) *T. metallica*, Tirol; (**f**) *T. corinthia*, Montenegro; (**d**–**f**) manubrium and flagellum, dorsal and lateral view.

**Figure 2 insects-12-00937-f002:**
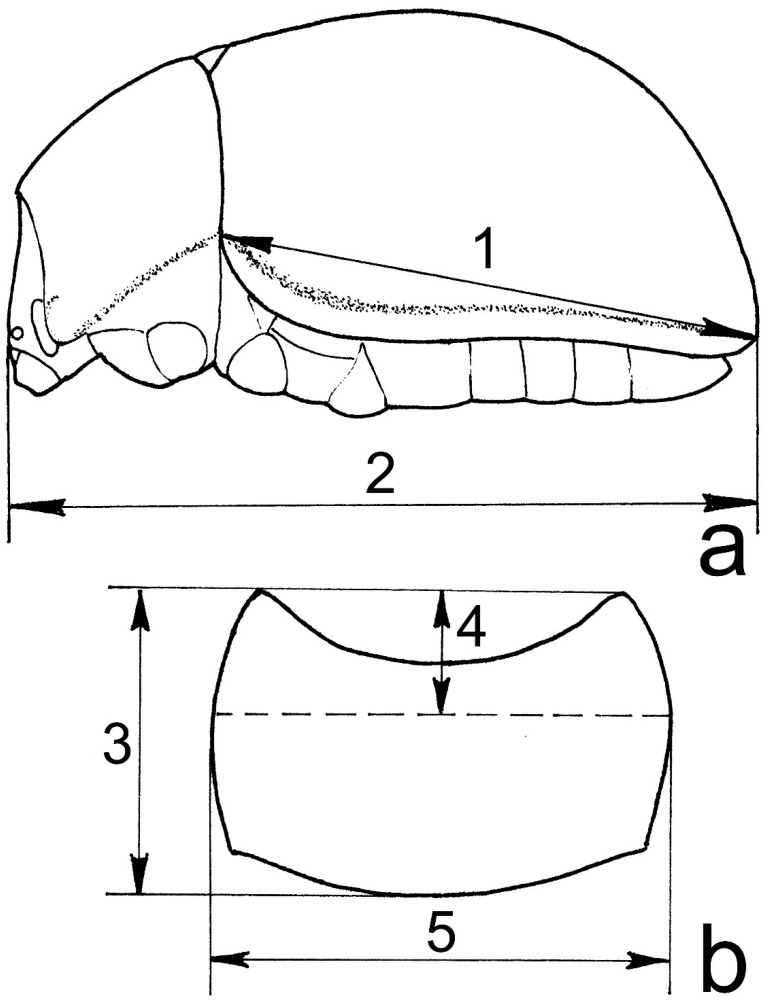
Examined morphological details. (**a**) General lateral view; (**b**) pronotum, dorsal view. Measured characteristics: 1—elytral length, 2—total body length, 3—pronotal length, 4— distance from anterior margin of the pronotum to maximal width, 5—pronotal width.

**Figure 3 insects-12-00937-f003:**
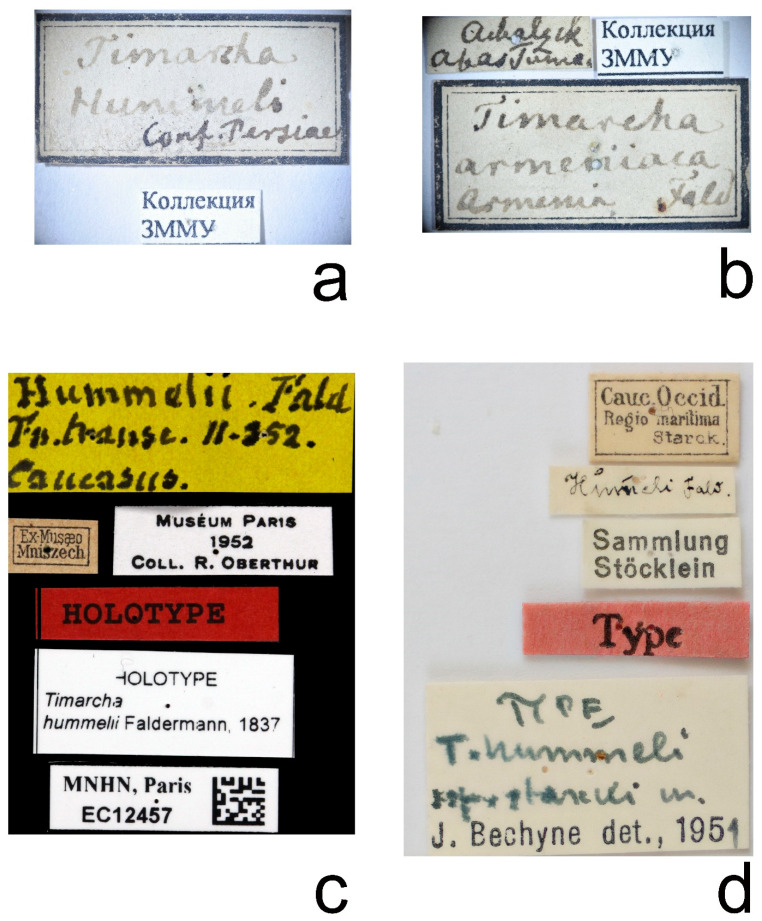
Labels of type and historical specimens. (**a**) *Timarcha hummelii*, neotype, male, ZMMU; (**b**) *T. armeniaca*, neotype, female, ZMMU; (**c**) *T. hummelii*, historical specimen, female, MNHN; (**d**) *T. hummelii starcki*, syntype, male, NMB.

**Figure 4 insects-12-00937-f004:**
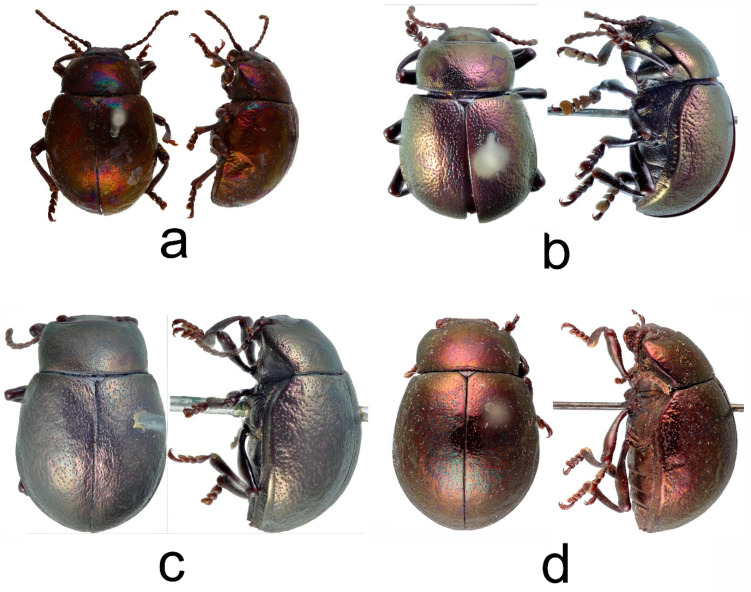
Type and historical specimens, dorsal and lateral view. (**a**) *Timarcha hummelii*, historical specimen, female, MNHN; (**b**) *T. hummelii*, neotype, male, ZMMU; (**c**) *T. armeniaca*, neotype, female, ZMMU; (**d**) *T. hummelii starcki*, syntype, male, NMB.

**Figure 5 insects-12-00937-f005:**
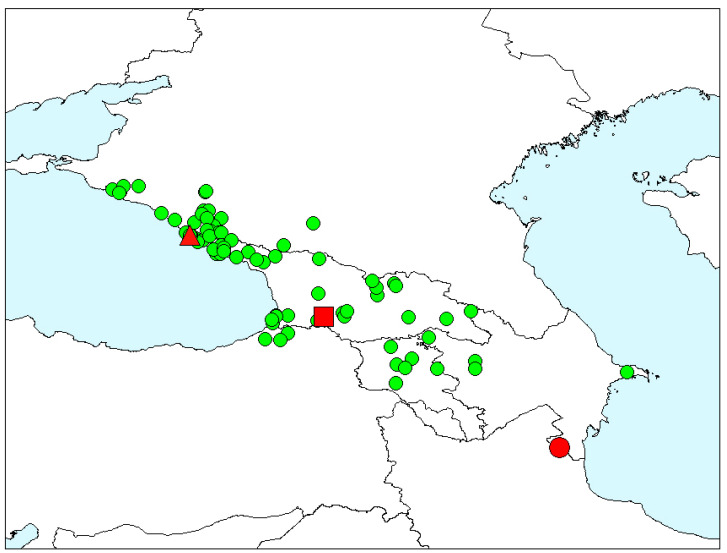
*Timarcha hummelii*, material examined. Red figures—type localities of the nominal taxa: triangle—*T. hummelii starcki*, square—*T. armeniaca*, circle—*T. hummelii*. Green circles—non-type specimens.

**Table 1 insects-12-00937-t001:** Material examined.

Region	Number of Males	Number of Females	Total
Western Caucasus, Russia	40	43	83
Western Caucasus, Abkhazia	5	15	20
Western Caucasus total	45	58	103
Сentral and southwestern Georgia, Northeastern Turkey	27	29	56
North and East Georgia, Armenia, Azerbaijan	18	17	35
Transcaucasia total	45	46	126
Western Europe	26	26	52
Balkans	6	9	15

**Table 2 insects-12-00937-t002:** Total body length, mm. M—mean value, σ—standard deviations, SE—standard error.

Taxon	Region Number	Region	Minimum, All Specimens	Maximum, All Specimens	M ± SE All Specimens	σ All Specimens
*hummelii*	1	Western Caucasus, Russia	6.86	10.98	8.89 ± 0.11	1.01
*hummelii*	2	Western Caucasus, Abkhazia	7.81	12.24	9.99 ± 0.27	1.22
*hummelii*	3	Western Caucasus total	6.86	12.24	9.10 ± 0.11	1.13
*hummelii*	4	Сentral and southwestern Georgia, Northeastern Turkey	7.39	12.46	9.79 ± 0.20	1.51
*hummelii*	5	North and East Georgia, Armenia, Azerbaijan	7.72	11.51	9.40 ± 0.19	1.10
*hummelii*	6	Transcaucasia total	7.39	12.46	9.64 ± 0.14	1.37
*metallica*	7	Western Europe	6.86	10.45	8.30 ± 0.12	0.86
*corinthia*	8	Balkans	9.08	12.78	10.86 ± 0.30	1.15

**Table 3 insects-12-00937-t003:** Total body length, mm. M—mean value, σ—standard deviations, SE—standard error.

Taxon	Region	Minimum, Females	Maximum, Females	M ± SE Females	σ Females	Minimum, Males	Maximum, Males	M ± SE Males	σ Males
*hummelii*	Western Caucasus, Russia	8.13	10.98	9.68 ± 0.10	0.63	6.86	9.50	8.03 ± 0.08	0.50
*hummelii*	Western Caucasus, Abkhazia	9.29	12.24	10.50 ± 0.22	0.87	7.81	9.61	8.44 ± 0.31	0.70
*hummelii*	Western Caucasus total	8.13	12.24	9.89 ± 0.10	0.78	6.86	9.61	8.08 ± 0.08	0.53
*hummelii*	Сentral and southwestern Georgia, NortheasternTurkey	9.50	12.46	11.09 ± 0.13	0.69	7.39	10.13	8.39 ± 0.12	0.63
*hummelii*	North and East Georgia, Armenia, Azerbaijan	9.71	11.51	10.38 ± 0.14	0.56	7.71	9.50	8.48 ± 0.12	0.51
*hummelii*	Transcaucasia total	9.50	12.46	10.82 ± 0.11	0.72	7.39	10.13	8.43 ± 0.09	0.58
*metallica*	Western Europe	7.60	10.45	8.86 ± 0.13	0.71	6.86	9.39	7.69 ± 0.10	0.53
*corinthia*	Balkans	9.92	12.77	11.38 ± 0.33	1.00	9.08	11.29	10.08 ± 0.39	0.96

**Table 4 insects-12-00937-t004:** Size of the pronotum (a): pronotal length (in dorsal view)/elytral length. M—mean value, σ—standard deviations, SE—standard error.

Taxon	Region Number	Region	Minimum	Maximum	M ± SE	σ
*hummelii*	1	Western Caucasus, Russia	0.41	0.62	0.50 ± 0.01	0.04
*hummelii*	2	Western Caucasus, Abkhazia	0.45	0.56	0.49 ± 0.01	0.03
*hummelii*	3	Western Caucasus total	0.41	0.62	0.50 ± 0.01	0.04
*hummelii*	4	Сentral and southwestern Georgia, Northeastern Turkey	0.41	0.59	0.50 ± 0.01	0.04
*hummelii*	5	North and East Georgia, Armenia, Azerbaijan	0.41	0.56	0.49 ± 0.01	0.03
*hummelii*	6	Transcaucasia total	0.41	0.59	0.49 ± 0.01	0.04
*metallica*	7	Western Europe	0.41	0.62	0.51 ± 0.01	0.04
*corinthia*	8	Balkans	0.43	0.49	0.46 ± 0.01	0.02

**Table 5 insects-12-00937-t005:** Size of the pronotum (b): pronotal width (in dorsal view)/elytral length. M—mean value, σ—standard deviations, SE—standard error.

Taxon	Region Number	Region	Minimum	Maximum	M ± SE	σ
*hummelii*	1	Western Caucasus, Russia	0.59	0.86	0.70 ± 0.01	0.06
*hummelii*	2	Western Caucasus, Abkhazia	0.60	0.77	0.67 ± 0.01	0.04
*hummelii*	3	Western Caucasus total	0.59	0.86	0.70 ± 0.01	0.05
*hummelii*	4	Сentral and southwestern Georgia, Northeastern Turkey	0.56	0.82	0.68 ± 0.01	0.06
*hummelii*	5	North and East Georgia, Armenia, Azerbaijan	0.60	0.82	0.68 ± 0.01	0.05
*hummelii*	6	Transcaucasia total	0.56	0.82	0.68 ± 0.01	0.06
*metallica*	7	Western Europe	0.59	0.84	0.71 ± 0.01	0.05
*corinthia*	8	Balkans	0.59	0.70	0.64 ± 0.01	0.03

**Table 6 insects-12-00937-t006:** Location of the maximal width of the pronotum: distance from the level of the front corners to the level of the greatest width of the pronotum/total length of the pronotum (both in dorsal view). M—mean value, σ—standard deviations, SE—standard error.

Taxon	Region Number	Region	Minimum	Maximum	M ± SE	σ
* hummelii *	1	Western Caucasus, Russia	0.20	0.55	0.38 ± 0.01	0.06
* hummelii *	2	Western Caucasus, Abkhazia	0.31	0.95	0.46 ± 0.03	0.13
* hummelii *	3	Western Caucasus total	0.20	0.95	0.40 ± 0.01	0.08
* hummelii *	4	Сentral and southwestern Georgia, Northeastern Turkey	0.25	0.80	0.44 ± 0.02	0.12
* hummelii *	5	North and East Georgia, Armenia, Azerbaijan	0.32	0.76	0.46 ± 0.02	0.09
* hummelii *	6	Transcaucasia total	0.25	0.80	0.45 ± 0.01	0.11
* metallica *	7	Western Europe	0.33	0.83	0.51 ± 0.02	0.13
* corinthia *	8	Balkans	0.39	0.56	0.46 ± 0.01	0.05

**Table 7 insects-12-00937-t007:** Emargination of pronotal lateral side before base (present or absent). RE—share representativeness error.

Taxon	Region Number	Region	Present% ± RE	Absent% ± RE
*hummelii*	1	Western Caucasus, Russia	83 ± 5	17 ± 5
*hummelii*	2	Western Caucasus, Abkhazia	70 ± 11	30 ± 11
*hummelii*	3	Western Caucasus total	81 ± 4	19 ± 4
*hummelii*	4	Сentral and southwestern Georgia, Northeastern Turkey	64 ± 6	36 ± 6
*hummelii*	5	North and East Georgia, Armenia, Azerbaijan	74 ± 8	26 ± 8
*hummelii*	6	Transcaucasia total	68 ± 5	32 ± 5
*metallica*	7	Western Europe	61 ± 7	39 ± 7
*corinthia*	8	Balkans	80 ± 10	20 ± 10

**Table 8 insects-12-00937-t008:** Dorsal color. RE—share representativeness error.

Taxon	Region Number	Region	(1) Violet% ± RE	(2) Blue% ± RE	(3) Golden Green% ± RE	(4) Golden Coppery% ± RE	(5) Bronze% ± RE	(6) Blackish Bronze% ± RE
*hummelii*	1	Western Caucasus, Russia	42 ± 5	2 ± 2	4 ± 2	52 ± 6	-	-
*hummelii*	2	Western Caucasus, Abkhazia	40 ± 11	10 ± 7	10 ± 7	40 ± 11	-	-
*hummelii*	3	Western Caucasus total	42 ± 5	4 ± 2	6 ± 2	48 ± 5	-	-
*hummelii*	4	Сentral and southwestern Georgia, Northeastern Turkey	45 ± 7	7 ± 3	8 ± 4	40 ± 7	-	-
*hummelii*	5	North and East Georgia, Armenia, Azerbaijan	54 ± 9	-	-	46 ± 9	-	-
*hummelii*	6	Transcaucasia total	48 ± 5	3 ± 2	5 ± 2	39 ± 5	-	-
*metallica*	7	WesternEurope	2 ± 2	4 ± 3	6 ± 3	2 ± 2	28 ± 6	58 ± 7
*corinthia*	8	Balkans	7 ± 7	20 ± 10	13 ± 9	-	-	78 ± 11

**Table 9 insects-12-00937-t009:** Color of femora. RE—share representativeness error.

Taxon	Region Number	Region	(1) Rufous% ± RE	(2) Piceous% ± RE	(3) Black% ± RE
*hummelii*	1	Western Caucasus, Russia	23 ± 5	51 ± 6	26 ± 5
*hummelii*	2	Western Caucasus, Abkhazia	35 ± 11	40 ± 11	25 ± 10
*hummelii*	3	Western Caucasus total	25 ± 4	49 ± 5	26 ± 4
*hummelii*	4	Сentral and southwestern Georgia, Northeastern Turkey	11 ± 4	62 ± 6	27 ± 6
*hummelii*	5	North and East Georgia, Armenia, Azerbaijan	17 ± 6	60 ± 8	23 ± 7
*hummelii*	6	Transcaucasia total	13 ± 4	62 ± 5	25 ± 5
*metallica*	7	Western Europe	37 ± 7	50 ± 7	13 ± 5
*corinthia*	8	Balkans	-	7 ± 7	93 ± 7

**Table 10 insects-12-00937-t010:** Color of tarsi. RE—share representativeness error.

Taxon	Region Number	Region	(1) Rufous% ± RE	(2) Piceous% ± RE	(3) Black% ± RE
*hummelii*	1	Western Caucasus, Russia	53 ± 5	40 ± 5	7 ± 3
*hummelii*	2	Western Caucasus, Abkhazia	55 ± 11	20 ± 9	25 ± 10
*hummelii*	3	Western Caucasus total	53 ± 5	36 ± 5	11 ± 5
*hummelii*	4	Сentral and southwestern Georgia, NortheasternTurkey	32 ± 6	57 ± 7	11 ± 4
*hummelii*	5	North and East Georgia, Armenia, Azerbaijan	49 ± 9	46 ± 6	5 ± 4
*hummelii*	6	Transcaucasia total	38 ± 5	53 ± 5	9 ± 3
*metallica*	7	Western Europe	13 ± 5	54 ± 7	33 ± 6
*corinthia*	8	Balkans	-	-	100

**Table 11 insects-12-00937-t011:** Punctures at the elytral disk. RE—share representativeness error.

Taxon	Region Number	Region	(State 1) % ± RE	(State 2) % ± RE	(State 3) % ± RE	(State 4) % ± RE	(State 5) % ± RE
*hummelii*	1	Western Caucasus, Russia	42 ± 5	30 ± 5	22 ± 5	6 ± 3	-
*hummelii*	2	Western Caucasus, Abkhazia	30 ± 11	35 ± 11	15 ± 8	20 ± 9	-
*hummelii*	3	Western Caucasus total	40 ± 5	31 ± 5	20 ± 4	9 ± 3	-
*hummelii*	4	Сentral and southwestern Georgia, Northeastern Turkey	4 ± 3	20 ± 5	20 ± 5	47 ± 7	9 ± 4
*hummelii*	5	North and East Georgia, Armenia, Azerbaijan	6 ± 4	34 ± 8	29 ± 8	31 ± 8	-
*hummelii*	6	Transcaucasia total	4 ± 2	25 ± 5	23 ± 4	43 ± 5	5 ± 2
*metallica*	7	Western Europe	9 ± 4	55 ± 7	17 ± 5	15 ± 5	4 ± 3
*corinthia*	8	Balkans	-	-	-	13 ± 9	87 ± 9

**Table 12 insects-12-00937-t012:** Border at upper margin of elytral epipleura near base. RE—share representativeness error.

Taxon	Region Number	Region	(1) Present% ± RE	(2) Absent% ± RE
*hummelii*	1	Western Caucasus, Russia	34 ± 5	66 ± 5
*hummelii*	2	Western Caucasus, Abkhazia	0	100
*hummelii*	3	Western Caucasus total	27 ± 4	73 ± 4
*hummelii*	4	Сentral and southwestern Georgia, Northeastern Turkey	0	100
*hummelii*	5	North and East Georgia, Armenia, Azerbaijan	0	100
*hummelii*	6	Transcaucasia total	0	100
*metallica*	7	Western Europe	91 ± 4	9 ± 4
*corinthia*	8	Balkans	93 ± 7	7 ± 7

**Table 13 insects-12-00937-t013:** Shine of elytron. RE—share representativeness error.

Taxon	Number of Region	Region	Shining Males % ± RE	Dull Males % ± RE	Shining Females % ± RE	Dull Females % ± RE
*hummelii*	1	Western Caucasus, Russia	80 ± 6	20 ± 6	35 ± 7	65 ± 7
*hummelii*	2	Western Caucasus, Abkhazia	80 ± 20	20 ± 20	33 ± 13	67 ± 13
*hummelii*	3	Western Caucasus total	80 ± 6	20 ± 6	34 ± 6	66 ± 6
*hummelii*	4	Сentral and southwestern Georgia, Northeastern Turkey	70 ± 9	30 ± 9	33 ± 13	67 ± 13
*hummelii*	5	North and East Georgia, Armenia, Azerbaijan	55 ± 12	45 ± 12	6 ± 6	94 ± 6
*hummelii*	6	Transcaucasia total	64 ± 7	36 ± 7	24 ± 6	76 ± 6
*metallica*	7	Western Europe	100	-	86 ± 7	14 ± 7
*corinthia*	8	Balkans	78 ± 17	22 ± 17	83 ± 15	17 ± 15

**Table 14 insects-12-00937-t014:** Shape of aedeagus apex in lateral view. RE—share representativeness error.

Taxon	Region Number	Region	(1) Recurved Dorsally% ± RE	(2) Evenly Curved% ± RE
*hummelii*	1	Western Caucasus, Russia	5 ± 2	95 ± 2
*hummelii*	2	Western Caucasus, Abkhazia	-	100
*hummelii*	3	Western Caucasus total	4 ± 3	96 ± 3
*hummelii*	4	Сentral and southwestern Georgia, Northeastern Turkey	-	100
*hummelii*	5	North and East Georgia, Armenia, Azerbaijan	-	100
*hummelii*	6	Transcaucasia total	-	100
*metallica*	7	Western Europe	81 ± 8	19 ± 8
*corinthia*	8	Balkans	-	100

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
