# Peer review of "The Structure of the Endophallus Is a New Promising Feature and a Key to Study of Taxonomy of the Subgenus Metallotimarcha of the Genus Timarcha (Coleoptera, Chrysomelidae) in the Caucasus"

_insects, 2021, doi:10.3390/insects12100937_

Round 1
Reviewer 1 Report
In the manuscript the taxonomic history of the species is clearly described, and no further information is rerquired.
However, the choice to test the taxonomic rank using the traditional morphometric methods is not the best one. The use of geometric morphometric method should have been a better choice, to evaluate size and shape variation.
also the colour morph definition could be improved using a objective method, as a colorimeter or a spectrometer.
Otherwise, the choice of the methods should be evaluated and described in details, and the manuscript should be nmodified
The description of the differences among subspecie in a bit complicated, perhaps a table would work.
Also the discussion part should be modified, according to the changes of the text.
Author Response
Response to Reviewer 1
Thank you for your thorough and friendly review.
- The choice to test the taxonomic rank using the traditional morphometric methods is not the best one. The use of geometric morphometric method should have been a better choice, to evaluate size and shape variation.
I agree with the reviewer that using the geometric morphometric method may be the best choice. However, I opted for the morphometric method, since the results obtained by this method can be more easily verified by subsequent scientists and more accessible for the study of new material (it is enough to have a stereomicroscope and a measuring eyepiece).
- The colour morph definition could be improved using a objective method, as a colorimeter or a spectrometer.
I agree with the reviewer that color detection can be improved with an objective method such as a colorimeter or spectrometer. However, I am unable to carry out this research, since most of the studied material was returned to museums in connection with the end of the borrowing period. The color differences described in the article are additional features, since it does not give 100% differences in all cases. On the other hand, I am not aware of any successful attempts to apply hardware-based color morph describing techniques in my field of ​​taxonomy, Chrysomelidae systematics. But I agree that this is a promising method for further research.
Otherwise, the choice of the methods should be evaluated and described in details, and the manuscript should be nmodified
I agree with you. The following text is added to the manuscript (chapter 2.4):
Objective methods for recording color morphs, such as colorimetry or spectrometry, may provide better results and may be the subject of future research. However, at present they are rarely used in taxonomic practice. The simple designation of colors with words prevails in the taxonomic literature and allows comparison of the results of different studies. In the case of the group under consideration, color morphs are an additional feature that does not give 100% differences.
The description of the differences among subspecie in a bit complicated, perhaps a table would work.
Besides the nominative subspecies Timarcha hummelii hummelii, only one subspecies, T. hummelii starcki has been previously described. The study of the material showed that T. hummelii starcki is invalid. It may not be necessary to compile a table with the characteristics of a non-existent subspecies? But if the reviewer sees fit, I could add it.
Also the discussion part should be modified, according to the changes of the text.
I do not know what could be changed in the Discussion, but I am ready to make the necessary changes if there are specific comments.
Reviewer 2 Report
This paper is well done and follows all the articles of the Code concerning the research of the type specimens and the designation of the neotypes adequately.
There is however an incongruity that the author has to explain: why he didn't designate a lectotype of T. hummelii starcki? It would have been important in a newly synonymized taxon.
Author Response
Reviewer: There is however an incongruity that the author has to explain: why he didn't designate a lectotype of T. hummelii starcki? It would have been important in a newly synonymized taxon.
Author's answer: Thank you very much for your review. I include an explanation in the text (chapter 3.1.2):
Tha author of the present study does not designate a lectotype. The designation of a lectotype is justified when there is reason to believe that a type series may include more than one taxon. As a result of the present study of syntypes, it was established that both of them belong to the same taxon. In this case, the selection of the lectotype is optional.
Reviewer 3 Report
The authors present a carefully done morphological study on the taxonomy of the coleopteran genus Timarcha, subgenus Metallotimarcha in the Caucasus, using field collected beetles as well as museum specimens. The manuscript is well structured and easy to read also for non-taxonomists. The scientific significance of the paper, however, is rather limited to morphologist / taxonomists. I still believe that the final decision on taxonomic issues should be done after molecular genetic studies. This is also shortly mentioned by the author of the paper, but such results are obviously not yet available. In this respect, the manuscript can be published in Insects after some minor revision:
- scientific manuscripts should not be written in the first person
- typing errors have to be corrected, such as in 3.2.1, 3.3.6 etc.
- I miss page numbers and line numbers in the manuscript
- insert a space between numbers and units, such as in 3.2.3
- all species names must be written in italics (e.g., see below tables and in 3.3.4, 3.3.5)
- grammatical errors (single, plural) as e.g. in 3.3.9
- use uppercase letters in headline 4
Author Response
Response to Reviewer 2
Thank you for your thorough and friendly review.
The final decision on taxonomic issues should be done after molecular genetic studies. This is also shortly mentioned by the author of the paper, but such results are obviously not yet available.
I agree with you. The following text is added to the manuscript (chapter 4):
The molecular genetic study of the subgenus Metallotimarcha may become the subject of further research and will allow testing the conclusions of this article at a new level of knowledge.
- scientific manuscripts should not be written in the first person
All first person is replaced in the manuscript by “the author of the present study”, “the author”, or passive voice (e.g. “It was studied...”), respectively in different cases.
- typing errors have to be corrected, such as in 3.2.1, 3.3.6 etc.
I corrected errors in 3.2.1. Unfortunately, I could not find an error in 3.3.6. Please, note this mistake aagain.
- I miss page numbers and line numbers in the manuscript
I don't understand why this is happening, but I am unable to add line numbers to this document. Is this a computer misunderstanding?
- insert a space between numbers and units, such as in 3.2.3 - I inserted the space in 3.2.3 and checked all other similsr places in the text.
- all species names must be written in italics (e.g., see below tables and in 3.3.4, 3.3.5) - I checked all the specific names, and made the necessary corrections (italics).
- grammatical errors (single, plural) as e.g. in 3.3.9 - I checked all the text and made corrections.
- use uppercase letters in headline 4 - Authors of “Insects” must use the Microsoft Word template. The headlines of the first level such as “ General discussion and conclusion” are not Uppercase.
This manuscript is a resubmission of an earlier submission. The following is a list of the peer review reports and author responses from that submission.